# Functional and Structural Brain Abnormalities and Clinical Characteristics of Male Patients with Alcohol Dependence

**DOI:** 10.3390/brainsci13060942

**Published:** 2023-06-12

**Authors:** Shin-Eui Park, Yeong-Jae Jeon, Hyeon-Man Baek

**Affiliations:** 1Lee Gil Ya Cancer & Diabetes Institute, Gachon University, Incheon 21999, Republic of Korea; shineuipark@gmail.com (S.-E.P.); yeong@gachon.ac.kr (Y.-J.J.); 2Department of Health Sciences & Technology, Gachon Advanced Institute for Health Sciences and Technology (GAIHST), Gachon University, Incheon 21999, Republic of Korea; 3College of Medicine, Gachon University, Incheon 21565, Republic of Korea

**Keywords:** male patients with alcohol dependence, alcoholism, white matter, globus pallidus, resting-state functional connectivity

## Abstract

Even though many previous studies have reported structural or functional brain abnormalities in patients with alcohol dependence (ADPs), studies observing the structural and functional abnormalities associated with the clinical characteristics of ADPs utilizing a multimodal approach are still scarce. The aim of this study was to demonstrate structural and functional brain abnormalities and their association with the clinical characteristics of alcoholism in male ADPs. Fifteen healthy male controls (HCs) and 15 male ADPs who had been diagnosed according to the Diagnostic and Statistical Manual of Mental Disorders 5 criteria underwent T1-weighted imaging and resting-state functional magnetic resonance imaging (MRI) scans. The MRI data were postprocessed using statistical parametric mapping for structural analysis and CONN-fMRI functional connectivity (FC) tools for functional analysis. In comparison with male HCs, male ADPs were characterized by significantly reduced volumes of the white matter in the left globus pallidus (GP) (*p*-FDR < 0.05). This region affected the altered resting-state FC patterns in male ADPs. Interestingly, an abnormal FC in the precuneus and its positive correlation with the alcohol-use disorder identification test score were observed in ADPs (r = 0.546, *p* = 0.036). Based on the observations, it could be concluded that the GP serves as a neural marker that impacts abnormal functional networks in men with alcohol dependence. These findings have important clinical implications as they provide insights into the neural mechanism underlying the anatomical, functional, and clinical features of alcoholism.

## 1. Introduction

Alcohol is a central nervous system depressant that can cause a range of neurological and cognitive problems, particularly when excessively consumed or over a long period [1]. Alcohol interferes with the normal functioning of neurotransmitters, which are responsible for transmitting signals between neurons in the brain. This can lead to decreased brain activity, impaired cognitive function, and memory deficits [2,3]. Moreover, chronic alcohol abuse can result in structural defects of the brain, including the loss of brain tissue, particularly in the prefrontal cortex and cerebellum [2,4]. The prefrontal cortex is involved in decision making, judgment, and impulse control; the cerebellum plays a critical role in coordination and balance. The loss of brain tissue in these areas can lead to a range of neurological symptoms, including tremors, loss of coordination, and speech impairment, and these effects may lead to social problems such as violence and crime in an affected individual [5]. Additionally, alcohol abuse and dependence can lead to numerous health conditions such as liver disease, cardiovascular problems, mental health disorders, and various cancers. Treating these conditions imposes a substantial burden on costs of healthcare systems [6].

Alcoholism can also trigger the development of a condition called Wernicke–Korsakoff syndrome, which is caused by a thiamine (vitamin B1) deficiency associated with chronic alcohol consumption. This condition can lead to severe cognitive impairment, memory deficits, and difficulty with coordination and balance [7]. 

Therefore, understanding the neural mechanisms related to alcoholism is of crucial importance in order to prevent long-term brain damage and cognitive impairment in patients with alcohol dependence (ADPs). To date, several attempts have been made to understand the neural mechanisms associated with alcoholism. They have relied on the neuroimaging approach that is widely used to investigate alcohol-induced neuronal abnormalities regarding brain structure [8,9,10] and function [11,12,13]. According to recent studies [14,15], structural abnormalities in the gray matter (GM) are linked to abnormal functional connections within neural networks in relevant brain regions in ADPs. These findings suggest that alterations in functional connectivity (FC) may be influenced by structural GM changes resulting from chronic alcoholism. However, to date, comprehensive multimodal neuroimaging studies examining brain changes in ADPs with a focus on both neurofunctional and structural abnormalities associated with clinical characteristics are scarce. Hence, the primary objective of our study was to provide a more comprehensive understanding of the neural underpinnings of alcoholism using multimodal approach.

Women exhibit lower levels of alcohol consumption, initiating their drinking habits at a later stage and displaying lower rates of alcohol dependence in comparison with men [16]. Nevertheless, research has indicated that women experience a more rapid progression from their initial alcohol use to developing alcohol dependence compared with men [17]. Consequently, given the presence of sex-related neuroanatomical variances as identified in a recent neuroimaging study [18], exclusively male subjects were chosen to account for these sex distinctions. For this reason, only male subjects were selected in this study.

Based on the information presented above, it was hypothesized that male ADPs have unique neuroanatomical features in the key brain regions that impact the functional networks. Characterizing such features may contribute to a better understanding of the differential pathophysiological mechanisms underlying alcoholism. To investigate these brain alterations, including structural abnormalities and their associations with FC changes in male ADPs, we adopted a multimodal approach utilizing voxel-based morphometry (VBM) and resting-state functional magnetic resonance imaging analyses. 

## 2. Materials and Methods 

Figure 1 provides an overview of data acquisition and processing.

### 2.1. Subjects

Our study recruited a total 34 participants, considering poor data quality. Among them, 15 healthy male controls (HCs; mean age, 47.60 ± 3.91 years) and 15 male ADPs (mean age, 47.53 ± 6.51 years) (Mann–Whitney U-test, *p* < 0.05), diagnosed by a psychiatrist based on the Diagnostic and Statistical Manual of Mental Disorders 5 (DSM-5) criteria, were included in this study (Table 1). All participants were right-handed (chi-square test, *p* = 1.000) and took the Brief Michigan Alcoholism Screening Test (BMAST) and Alcohol Use Disorder Identification Test-Korea (AUDIT-K). BMAST and AUDIT-K, which consist of 10 questions, are self-administered, with individuals responding to the questionnaire. Each question has a scoring system, and the total score is calculated by summing up the scores from each question. Higher scores indicate a higher likelihood of alcohol-related problems or alcohol-use disorders. Before undergoing MRI, all volunteers received a thorough explanation of all experimental procedures and each provided written informed consent. Participants in both groups showed no abnormalities on physical or neurological examination. Additionally, all the experimental procedures and methods were performed in accordance with the relevant guidelines and regulations approved by the IRB-CBUH (Chunbuk National University Hospital) and participants underwent MRI study in Korea Basic Science Institute (KBSI).

#### 2.1.1. Inclusion Criteria

DSM-5 criteria for ADPs.Male patients between 20 and 60 years of age.All participants had an education above middle school (more than 9 years: middle school in Korea).Right-handed.

#### 2.1.2. Exclusion Criteria

Current or past psychiatric illness other than ADPs—dementia, delirium, and other organic disorders; mental retardation; psychotic disorders such as schizophrenia, delusional disorder, and others; mood disorders; and anxiety disorders.Substance use other than alcohol and tobacco.History of head trauma.History of cardiovascular or endocrine disease.Current medical illness.Presence of magnetically active object in the body.

### 2.2. MRI Acquisition

The functional and structural data were collected by a 3 Tesla MRI system (Philips Achieva scanner, Philips Healthcare, Best, the Netherlands) with an 8-channel head coil. During the scan, patients were in a supine position with their heads supported and immobilized within the head coil with foam-pads (vendor-provided) to minimize head movement and gradient noise.

The high-resolution magnetic resonance images were obtained with a three-dimensional Turbo field echo sequence. The acquisition parameters were as follows: repetition time (TR) = 6.8 ms, echo time (TE) = 3.1 ms, flip angle = 9°, field-of-view (FOV) = 288 × 288 cm^2^, matrix size = 256 × 256, number of excitation (NEX) = 1, slice thickness = 1.2 mm, and number of slices = 170.

Functional images were acquired using a gradient-echo echo planner image sequence with the following parameters: TR/TE = 3000/30 ms, flip angle = 80°, FOV = 22 × 22 cm^2^, matrix size = 64 × 64, NEX = 1, number of slices = 48, isotropic voxels = 3 mm, and total scan time = 7 min. 

These parameters used in the MRI acquisition followed those in the procedure described in our previous study [19].

### 2.3. Brain Volume Difference

The high-resolution T1 images were post-processed using statistical parametric mapping (SPM12) software with diffeomorphic anatomical registration through the exponentiated lie-algebra-based VBM analysis following the procedure described in our previous study [20]. Alternating GM and white matter (WM) volumes in both groups were assessed with an independent two-sample *t*-test with multiple comparisons using family-wise error (FWE) at *p* < 0.05 and cluster size > 100 voxels.

### 2.4. FC Analysis

The FC analysis was performed using the CONN-fMRI FC toolbox (ver. 21a) with SPM12 software. The preprocessing was performed as follows: slice timing correction (interleaved), realignment of images (subject-motion threshold: 2 mm), and filed map correction; individual T1 images were coregistered with functional images. The transformed T1 images were segmented to GM, WM, or cerebrospinal fluid using standard SPM tissue probability maps. Next, these images underwent spatial smoothing with full width at half maximum Gaussian kernel at 8 mm^3^. For optimal functional outlier detection, the liberal setting with 97th percentiles in the normative sample and a global-signal z-value threshold of 9 was adopted. To minimize the influence of noise on the blood oxygenation level dependent (BOLD) signals, such as subject motion, the identified outlier scans or scrubbing artifacts [21] were reduced with an anatomical component-based noise correction method (aCompCor) [22]. Band-pass filtering was performed with a frequency window of 0.01–0.1 Hz. 

For the evaluation of FC, a region of interest (ROI)-to-voxel FC analytical approach was used. Referring to the VBM results, we defined an ROI at the left globus pallidus (GP) (MNI coordinate: x = −21, y = 2, z = 0, as a 10 mm sphere). Significant connections were identified with voxel threshold calculations: uncorr. *p* < 0.001, cluster threshold: false discovery rate (FDR) at *p* < 0.05.

### 2.5. Statistical Analysis

The mean age of participants was examined through the utilization of the Mann–Whitney U-test. The variables of hand dominance and sex were assessed employing the chi-square test.

To analyze the correlation between FC intensities and alcoholism severity (BMAST and AUDIT-K scores), Pearson’s correlation coefficient test was performed using SPSS Statistical software package (version 20.0, SPSS Inc., Chicago, IL, USA).

## 3. Results

### 3.1. Demographic and Clinical Measurements

Table 1 shows demographic and clinical characteristics of the ADPs and HCs. No differences between the two groups were noted in terms of age, handedness, and sex distribution; however, there were significant differences in the levels of education and durations of cigarette smoking. The days of alcohol consumption per week were 4.73 ± 2.05 (mean ± SD) and 1.57 ± 1.79 days for ADPs and HCs, respectively. 

**Table 1 brainsci-13-00942-t001:** Demographic and clinical characteristics of male patients with alcohol dependence (ADPs) and healthy controls (HCs).

	Alcohol-Depent Patients (n = 15)	Healthy Controls (n = 15)	*p*-Value
Mean age (years)	47.53 ± 6.51	47.60 ± 3.91	*p* = 0.902 *
Sex (male/female)	15/0	15/0	*p* = 1.000 **
Handedness (right: left: mixed)	15:0:0	15:0:0	*p* = 1.000 **
Education (years)	12.27 ± 2.71	14.53 ± 2.20	*p* = 0.023 *
Duration of smoking (years)	26.87 ± 9.25	16.87 ± 12.69	*p* = 0.015 *
** * Clinical rating scales * **			
Duration of drinking per week (days)	4.73 ± 2.05	1.57 ± 1.79	*p* < 0.000 *
^a^BMAST	22.27 ± 5.36	2.07 ± 3.59	*p* < 0.000 *
^b^AUDIT-K	31.47 ± 7.17	11.87 ± 6.83	*p* < 0.000 *

* Mann–Whitney U-test, ** chi-square test, ^a^BMAST: Brief Michigan Alcoholism Screening Test, ^b^AUDIT-K: Alcohol Use Disorder Identification Test-Korea.

Regarding the alcoholism tests (Table 1), ADPs scored significantly higher on both the BMAST and AUDIT-K than HCs (*p* < 0.000). The mean scores for the BMAST and AUDIT-K were 22.27 ± 2.71 and 31.47 ± 7.17 for ADPs and 2.07 ± 3.59 and 11.87 ± 6.83 for HCs, respectively.

### 3.2. Brain Volume Difference

There was no significant difference in the GM between the two groups. In the WM, ADPs showed reduced volumes in the GP (two-sample *t*-test, *p*-FWE < 0.05, and cluster size > 100 voxel)s (Figure 2). 

### 3.3. Differences in the FC

The FC patterns derived from the GP between ADPs and HCs are shown in Figure 3 and Table 2. Although both groups were characterized by similar patterns, there were weaker and more restricted interregional connections in ADPs than in HCs. ADPs showed relatively higher FCs with the GP in the right putamen (Pu) and both right and left (right/left) cerebellum (Cb). In comparison, regions of the right/left middle frontal gyrus (MFG), right inferior frontal gyrus (IFG), right precentral gyrus (PreCG), left supplementary motor area (SMA), left supramarginal gyrus (SMG), right/left middle cingulate gyrus (MCG), and right superior temporal gyrus (STG) were observed in the HCs. The between-group contrast revealed a significantly lower FC in the precuneus (PCUN) in the ADPs than in the HCs (two-sample *t*-test, voxel threshold: uncorr. *p* < 0.001, and cluster threshold: *p*-FDR < 0.05) (Figure 4). 

### 3.4. Correlation between FC and Degree of Alcoholism 

Figure 5 shows a significant positive correlation between AUDIT-K scores and the FCs of the GP and PCUN with reduced local WM volumes in male ADPs (Pearson’s correlation (r) = 0.546, *p* = 0.036). However, there was no significant correlation with the BMAST scores.

## 4. Discussion

In the present study, we employed a comprehensive whole-brain analysis to delineate the structural and functional abnormalities in the brains of male ADPs and subsequently examined their association with alcoholism severity. The aim of this study was to identify the most vulnerable brain regions in alcoholism via qualitative and quantitative approaches. Our primary objective entailed two key components. Firstly, we aimed to identify the specific brain regions affected by the structural abnormalities attributed to alcoholism. Subsequently, we established a foundation for conducting functional analysis within these identified regions. Secondly, our focus was observing and analyzing the patterns of FC between two distinct groups. Ultimately, our aim was to present compelling evidence linking brain functional and/or structural abnormalities to the clinical characteristics exhibited by male ADPs. In the group comparison of brain volumes, there were no significant differences between the two groups in the GM. However, the ADPs exhibited significantly reduced WM volumes in the left GP.

Although there is conflicting evidence on whether light-to-moderate alcohol consumption has similar negative implications for the brain structure, it is widely accepted that alcohol consumption may cause brain atrophy, neuronal loss, and impaired WM fiber integrity [23]. The GP is part of the basal ganglia, a group of structures in the brain that are involved in motor control, reward processing, and learning [24,25]. In addition, a couple of studies [26,27] have highlighted the presence of GP structural abnormalities in ADPs. In particular, Zahr et al. [26] suggested that ADPs may have GP abnormalities linked to flawed reward processing and decision making.

According to a previous study, which reported that damaged neuronal cells may cause loss of brain function [28], the GP was established as a seed area to assess FC differences. In the evaluation of brain FC between male ADPs and HCs, the overall patterns of brain FC were similar between the two groups; however, some differences were noted. ADPs showed weaker and more restricted inter-regional activation synchrony within the connectivity network than the HCs. This suggested that the communication between the GP and different brain regions in the ADPs was less synchronized compared with that in the HCs.

In particular, ADPs showed relatively higher FCs with the GP in the right Pu, and right and left Cb were characterized by the highest BOLD signal in each cluster. A few functional MRI studies have investigated alcohol dependence and reported differential functional brain activities in the Pu and Cb. Sjoerds et al. [29] reported that ADPs showed decreased brain activation in Pu compared with that in HCs during an instrumental learning task designed to study the balance between goal direction and habit learning. Chanraud et al. [30] found that ADPs showed a lower FC between the posterior cingulate regions and Cb in resting state but a higher FC during a working memory task. These regions were associated with cognitive function and alcoholism. 

In comparison, the regions of right/left MFG, right IFG, right PreCG, left SMA, left SMG, right/left MCG, and right STG were more commonly observed in the HCs than in the ADPs. The GP is a key region in the cortico–striato–thalamo–cortical (CSTC) circuit. This circuit comprises well-integrated regulatory pathways that link the cortical and subcortical regions, which are involved in cognition, attentional control, motivation, motor control, and salience. Abnormal functional activation in the CSTC circuit may be a trigger for some psychiatric disorders [31,32]. Importantly, the observed brain regions in the HCs in our study were involved in CSTC pathways. Alcohol dependence can indeed have an impact on the vestibular system, which is responsible for maintaining our sense of balance and spatial orientation [33]. When the vestibular system is affected by alcohol abuse, it can lead to a condition known as alcoholic or alcohol-induced vestibular dysfunction [33]. The neuronal connections of the vestibular system extend from the peripheral receptors to various associated brain regions involved in processing vestibular information [34]. Among the regions observed in the FC results, Rt/Lt Cb, Rt/Lt MFG, and Lt. SMA and Lt. SMG are known to be directly or indirectly related to the vestibular system. Although not typically considered as primary components of the vestibular system, they are important brain regions involved in various cognitive and motor functions.

In the group comparison analysis, the ADPs showed significantly lower FCs in the PCUN than the HCs. Interestingly, these decreased FCs of the GP-PCUN had a positive correlation with the AUDIT scores in the ADPs. Yuan et al. [35] investigated the resting-state connectivity alterations associated with chronic heroin use; however, the study also included a group of individuals with alcohol dependence. The authors reported that both heroin and alcohol dependence were associated with altered small-world brain functional networks in the PCUN. Thus far, several studies [36,37,38] have reported abnormal FCs of the GP-PCUN in various mental disorders. In addition to these findings, our study contributes to the literature by providing additional evidence of abnormal FCs and their correlation with clinical severity in male ADPs. To the best of our knowledge, this is the first study analyzing the association between abnormal FCs and alcohol dependence. Considered collectively, an abnormal FC of the GP-PCUN in male ADPs can have significant consequences for various aspects of their lives. These effects can range from cognitive deficits and impaired self-awareness to increased craving for alcohol, motor impairments, and emotional dysregulation. Understanding these neurobiological changes could help in developing targeted interventions and treatment strategies for individuals struggling with alcoholism.

The current study has certain limitations. First, the sample size was small, consisting of only 15 male ADPs and 15 male HCs. Therefore, our findings may not be generalizable to larger populations or women. Future studies on larger samples and appropriate sex-matched control will be needed to increase the statistical significance and explore the generalizability of our results to other populations. Second, potential confounding effects of smoking and education levels on the brain structure and function could not be completely excluded. However, to minimize their influence, we considered these factors as covariates during imaging processing. Nonetheless, the dominance of alcoholism in contributing to the differences in the brain abnormalities between ADPs and HCs is consistent with the results of previous reports [14,39].

Overall, while this study provides valuable insights into the neural mechanisms underlying alcohol dependence, the limitations should be taken into consideration when interpreting the results. Future research should aim to address these limitations to provide a more comprehensive understanding of the effects of alcoholism on brain structure and function.

## 5. Conclusions

The present study found a significant reduction in the WM volume in the GP among male ADPs, and this reduction was associated with changes in the FC. These findings suggest that alcoholism can lead to structural and functional network alterations in key brain regions, providing insights into the neural mechanisms underlying male ADPs and potential implications for future treatment research.

## Figures and Tables

**Figure 1 brainsci-13-00942-f001:**
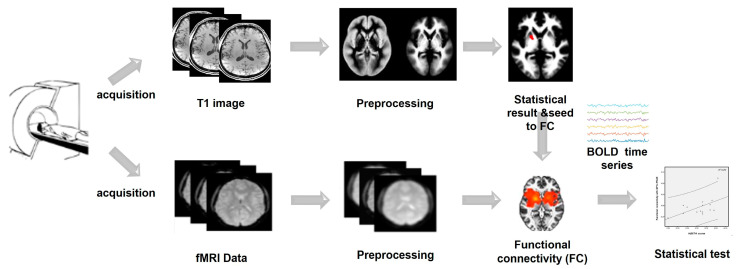
Illustration of data acquisition and processing.

**Figure 2 brainsci-13-00942-f002:**
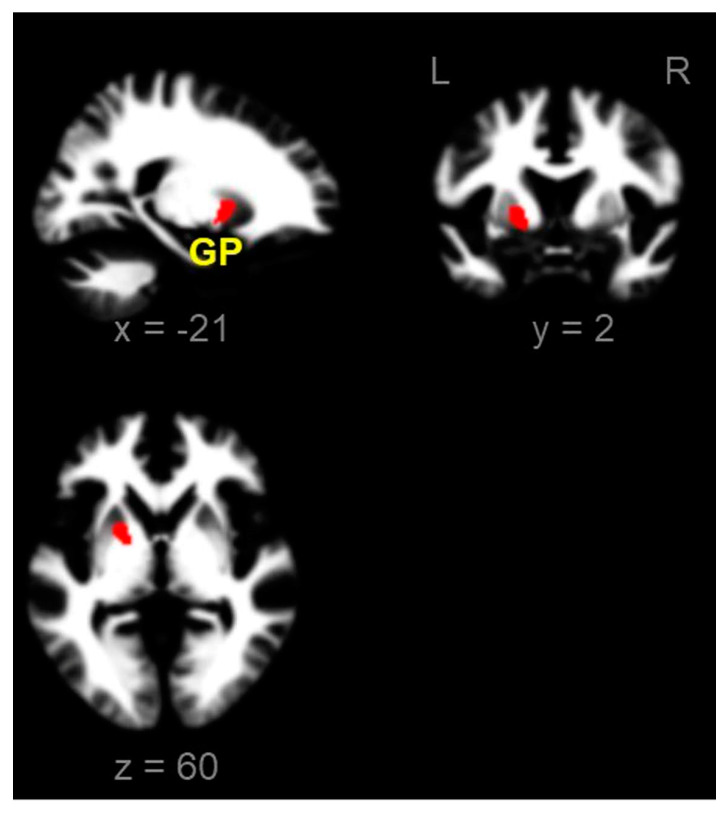
Reduced local white matter volumes in alcohol-dependent patients compared with those in healthy controls (two sample *t*-test, *p*-FWE < 0.05, and cluster size > 100 voxels). L, left; R, right; GP, globus pallidus (x, y, z = −21, 2, 0); FWE, family-wise error.

**Figure 3 brainsci-13-00942-f003:**
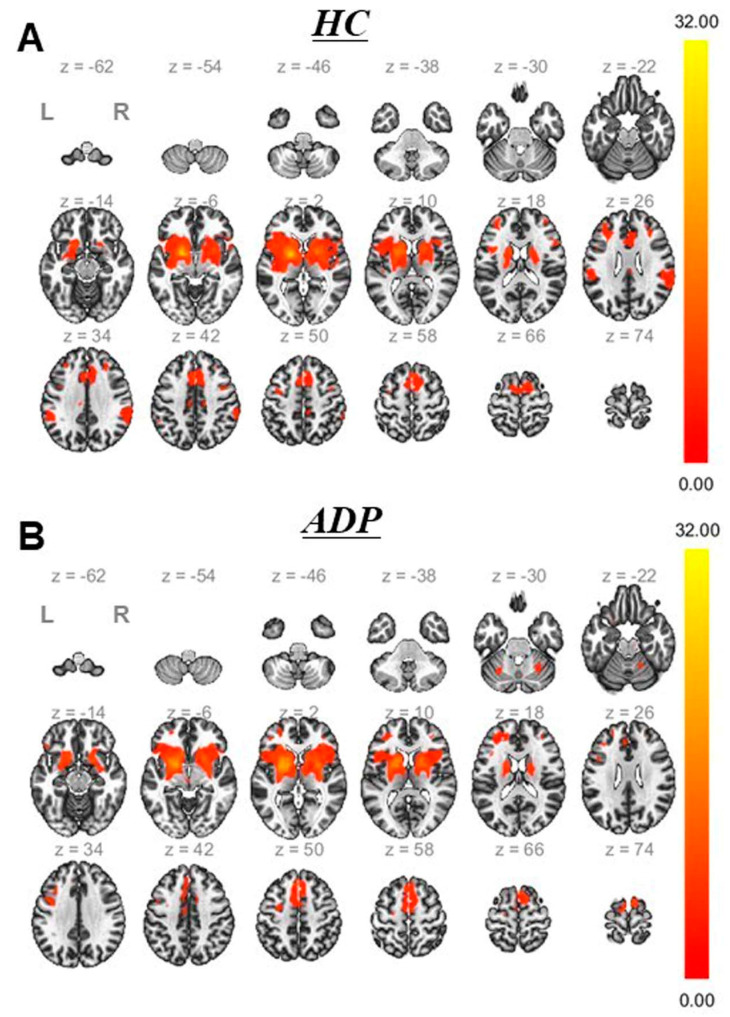
Positive connectivity network linked with altered white matter volumes in the globus pallidus in healthy controls (**A**) and alcohol-dependent patients (ADPs) (**B**). (One-sample *t*-test, voxel threshold uncorr. *p* < 0.001, and cluster threshold *p*-FDR < 0.05.) L, left; R, right; FDR, false discovery rate.

**Figure 4 brainsci-13-00942-f004:**
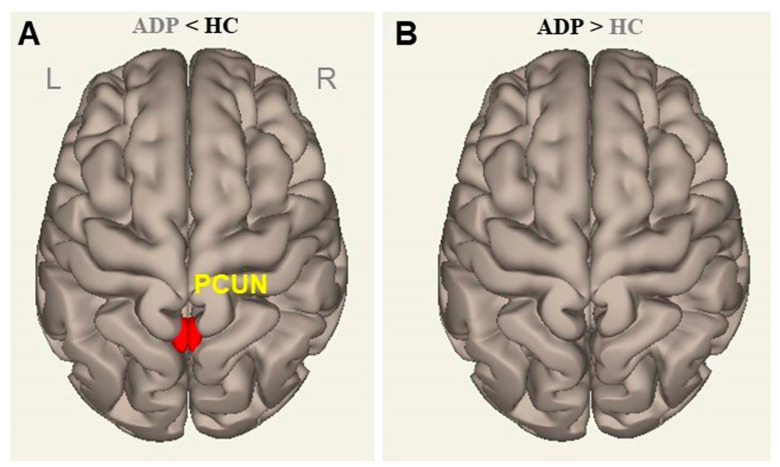
Distinct functional connectivity areas linked to altered white matter volumes (globus pallidus; x, y, z = −21, 2, 0) between ADPs and HCs (two-sample *t*-test, voxel threshold: uncorr. *p* < 0.001, and cluster threshold: *p*-FDR < 0.05). (**A**) ADP < HC; (**B**) ADP > HC; L, left; R, right; ADPs, patients with alcohol dependence; HCs, healthy controls; PCUN, precuneus; FDR, false discovery rate.

**Figure 5 brainsci-13-00942-f005:**
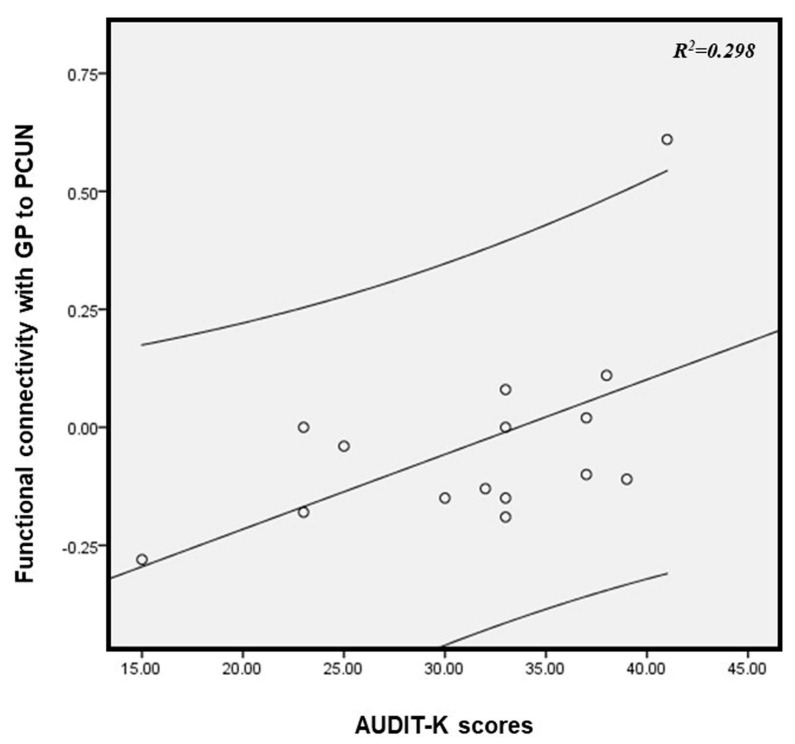
The correlation between AUDIT-K scores and functional connectivity of the precuneus (PCUN) linked to the globus pallidus (GP) with reduced white matter volume in alcohol-dependent patients (r = 0.546, *p* = 0.036). The curved line bands indicate 95% confidence intervals. AUDIT-K, Alcohol Use Disorders Identification Test-Korea.

**Table 2 brainsci-13-00942-t002:** Regions showing increased functional connectivity linked to the globus pallidus (GP) in patients with alcohol dependence and healthy controls (voxel threshold: uncorr. *p* < 0.001, and cluster threshold: *p*-FDR < 0.05).

Brain Area	MNI Coordinates			Cluster Size	Maximum *t*-Value
	x	y	z		
Seed: Globus pallidus	−21	2	0		
Patients with alcohol dependence					
Right putamen (Rt. Pu)	28	2	2	5491	11.34
Right cerebellum (Rt. Cb)	30	−60	−28	377	5.79
Left cerebellum (Lt. Cb)	−30	−62	−30	151	5.42
Healthy controls					
Right middle frontal gyrus (Rt. MFG)	26	32	38	506	5.11
Left middle frontal gyrus (Lt. MFG)	−32	54	20	701	6.72
Right inferior frontal gyrus (Rt. IFG)	56	18	18	171	4.56
Right precentral gyrus (Rt. PreCG)	40	0	46	200	4.79
Left supplementary motor area (Lt. SMA)	−4	14	54	3417	6.40
Left supramarginal gyrus (Lt. SMG)	−52	−42	36	683	5.45
Right middle cingulate gyrus (Rt. MCG)	2	−36	46	274	7.36
Left middle cingulate gyrus (Lt. MCG)	−10	−24	40	146	7.23
Right superior temporal gyrus (Rt. STG)	62	−50	24	1360	5.82

FDR: False discovery rate, MNI: Montreal neurologic institute.

## Data Availability

Data are unavailable due to privacy or ethical restrictions.

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
