# Peer review of "Functional and Structural Brain Abnormalities and Clinical Characteristics of Male Patients with Alcohol Dependence"

_brainsci, 2023, doi:10.3390/brainsci13060942_

Round 1

Reviewer 1 Report

Comments and Suggestions for Authors

Thanks for the opportunity to review this manuscript, my thoughts are below:

Abstract

- In this excerpt from the abstract “studies that have observed structural and functional abnormalities associated with clinical characteristics of ADPs utilizing a multimodal approach are still scarce”, the authors imply that they will make an association between functional and structural abnormalities with clinical characteristics of the male volunteers who are dependent on alcohol. If so, the title of the study needs to be better reformulated.

- What is the purpose of the study? The precise objective is in the abstract.

- What is the type/design of the study? The precise type/design of the study is in the abstract.

- What would this T1 be in the abstract? This term needs to be made clearer to readers.

- The results do not bring p-values, so it is difficult to know which outcomes showed only clinical differences and in which outcomes the differences were statistical between groups. I suggest including the p-value of the outcomes and using only the main outcomes in the abstract, so as not to confuse the reader and then, in the results section of the article, the authors can better explain all the analyzed outcomes.

- In addition, it is difficult to know which outcomes were primary and secondary, I suggest including in the abstract only the primary outcomes, the secondary outcomes the authors can mention in the results section of the article.

- Without knowing the objective of the study, it is difficult to know whether or not the conclusion is coherent and whether it responds to the objective of the study. However, the conclusion is a little superficial and not directed towards a clinical conclusion, I suggest improving the conclusion.

Introduction

- In the introduction the authors state that: “However, to date, there is scarcity of comprehensive multimodal neuroimaging studies examining brain changes in ADPs with a focus on both neurofunctional and structural abnormalities”, however, in the abstract they state that this information is scarce related the clinical features in ADPs. This excerpt was confusing when compared to the text of the abstract.

- In addition, from the introduction, it was not clear whether the authors intend to elucidate this gap in the literature on the topic, is that right?

- Where is the purpose of the study at the end of the introduction?

Materials and methods

- Include the type/design of the study.

- Where are the study eligibility criteria? What were the criteria for being included or excluded from the study? This information needs to be included in the article, please include it.

- How were volunteers recruited? Where were the volunteers recruited from? How did the authors arrive at this number of 30 volunteers? Was there any calculation to estimate the sample size, or was the study sample estimated by convenience? This information needs to be included in the article, please include it.

- The authors mention that they used two instruments to assess the volunteers: Brief Michigan Alcoholism Screening Test (BMAST) and Alcohol Use Disorder Identification Test-Korea (AUDIT-K), however, they do not provide further information about this assessment, that is, they do not inform how they used this instrument, what these instruments measure, how these measurements are performed (clinical tests, or by questions, functional tests), what are the scores of these instruments? However, they use these data in Table 1, using means and standard deviation. The authors should better explain each of the instruments, explaining how each instrument was used and measures the outcomes, which outcomes are evaluated by these instruments and how the score or punctuation of each of these instruments used by the authors occurs, so that the reader understands better the whole method of this study.

- Item 2.2 MRI Acquisition could have a picture of this acquisition, for a better understanding of how this evaluation was carried out. I suggest including.

- Were the parameters used in the MRI acquisition, Brain Volume Difference and FC Analysis based on parameters from previous studies? If yes, include references;

- The authors used Pearson's Chi-Square test and the Mann-Whitney U-test, however, analyzes using these tests were not mentioned in the statistical analysis section of the article, please include.

- The authors mention that they will make a correlation between (FC intensities and alcoholism severity scores), however, in the methods the authors do not mention which of the instruments used measures the severity of alcoholism, even though these instruments are widely used and known worldwide, the authors need to explain better which instrument was used for which outcome in the methods, so that it makes sense to the readers.

Results

- Were the ADP volunteers in this study light, moderate or heavy drinkers? This information needs to be clear in the text, not only in this group, but also in the control group.

Discussion

- Only in the discussion does the objective of the study appear, however, it seems incomplete, given the many evaluations that the authors carried out. I suggest authors to better delineate the objectives of the study, using terminologies such as: primary objective, secondary objective, tertiary... in this way, the text becomes clearer for readers.

- In the second paragraph of the discussion, the authors mention that “several studies have highlighted...” and then, they mention only two studies, or the volunteers change the term several, or add more studies, because the term “several studies” from the idea of “many studies” and only two studies seem not to refer to the term used. I suggest modifying.

- The authors mention some similar studies, however, they do not describe the characteristics of these studies, such as the sample size. Is the sample of these studies larger or smaller than that of the present study? This information can guide the reader as to which results have greater statistical power and also to determine the importance of the present study, when citing studies with a smaller sample size and similar methodology.

- The ADP's volunteers were smokers and had a lower level of education, can such outcomes influence the results? What is in the literature about the influence of these two outcomes on the evaluated brain components? I strongly suggest the construction of one or two paragraphs demonstrating the influence of these two outcomes on the evaluated brain characteristics.

- I also missed a paragraph demonstrating a relationship between: what the authors found and its repercussions for the lives of volunteers. Findings in these brain areas can lead to damage in which functional aspects of the volunteers? This information is important, as it can give rise to new research on the subject and also the implementation of intervention programs to benefit the studied population, I strongly suggest the inclusion of this paragraph.

- I also believe that the authors could mention some kind of influence of the chronic use of alcohol on the vestibular system, since they mention the balance problems caused by the continuous use of alcohol on body balance in the introduction. This citation is important, as future studies may include the vestibular components (peripheral and/or central) in future studies on the subject.

Conclusion

- The authors can leave the conclusion more direct to what the authors found.

Author Response

Abstract

- In this excerpt from the abstract “studies that have observed structural and functional abnormalities associated with clinical characteristics of ADPs utilizing a multimodal approach are still scarce”, the authors imply that they will make an association between functional and structural abnormalities with clinical characteristics of the male volunteers who are dependent on alcohol. If so, the title of the study needs to be better reformulated.

> As your advice, we revise the title like this : “ The Brain functional and structural abnormalities with clinical characteristics of male patients with alcohol dependence ”.

Refer to Title page in lines 2-5 on page 1.

- What is the purpose of the study? The precise objective is in the abstract.

> As your advice, we revise the abstract for more precise objective.

Refer to Abstract section, (lines 14-16 on page 1).

- What is the type/design of the study? The precise type/design of the study is in the abstract.

> Our paper is research article so we change the type/design from “Original Research” to “Article”. And corrected the Abstract structure as removing sub-title such as “Objectives” , “Methods” etc.

Refer to Abstract section, on page 1.

- What would this T1 be in the abstract? This term needs to be made clearer to readers.

> In MRI (Magnetic Resonance Imaging) physics, T1 refers to the longitudinal relaxation time, also known as spin-lattice relaxation time. It is a fundamental property of tissues and is related to the recovery of the magnetization along the direction of the applied magnetic field. T1 is intrinsic term and we can’t describe more precisely.

- The results do not bring p-values, so it is difficult to know which outcomes showed only clinical differences and in which outcomes the differences were statistical between groups. I suggest including the p-value of the outcomes and using only the main outcomes in the abstract, so as not to confuse the reader and then, in the results section of the article, the authors can better explain all the analyzed outcomes.

> As your suggestion, we include the P-value and reduced the results concisely in Abstract section.

Refer to Abstract section (in lines 22-27 on page 1).

- In addition, it is difficult to know which outcomes were primary and secondary, I suggest including in the abstract only the primary outcomes, the secondary outcomes the authors can mention in the results section of the article.

> As your suggestion, we eliminated the secondary outcomes in Abstract section.

Refer to Abstract section (in line 23-27 on page1).

- Without knowing the objective of the study, it is difficult to know whether or not the conclusion is coherent and whether it responds to the objective of the study. However, the conclusion is a little superficial and not directed towards a clinical conclusion, I suggest improving the conclusion.

> As your suggestion, we revised the conclusion like this : “Based on the observations, it can be concluded that the GP serves as a neural marker that impacts abnormal functional networks in males with ADPs. These findings have important clinical implications, as they provide insights into the neural mechanism underlying the anatomical, functional, and clinical features of alcoholism.”

Refer to Abstract section (in lines 29-35 on page 1).

Introduction

- In the introduction the authors state that: “However, to date, there is scarcity of comprehensive multimodal neuroimaging studies examining brain changes in ADPs with a focus on both neurofunctional and structural abnormalities”, however, in the abstract they state that this information is scarce related the clinical features in ADPs. This excerpt was confusing when compared to the text of the abstract.

> As your advice, we mentioned about “~ abnormalities associated with clinical characteristics” in Introduction section (Refer to line 70-71 on page 2).

- In addition, from the introduction, it was not clear whether the authors intend to elucidate this gap in the literature on the topic, is that right?

> Yes. We added the sentence of “Hence, the primary objective of our study is to provide a more comprehensive understanding of the neural underpinnings of alcoholism using multi modal approach.” to make our intention more clearly known in Introduction section (Refer to lines 71-73 on page 2).

> What is the difference between previous studies is multi modal approach.

- Where is the purpose of the study at the end of the introduction?

> We added the purpose of our study in Introduction section. This is same sentence of  above mentioned : “ Hence, the primary objective of our study is to provide a more comprehensive understanding of the neural underpinnings of alcoholism using multi modal approach “ . (Refer to lines 71-73 on page 2).

Materials and methods

- Include the type/design of the study.

> As your advice, we corrected the type and design.

“MATERIALS AND METHODS” to “2. Materials and Methods” , with sub-titles.

 (Refer to section of Materials and Methods).

- Where are the study eligibility criteria? What were the criteria for being included or excluded from the study? This information needs to be included in the article, please include it.

> As your advice, we added the criteria for inclusion and exclusion

 (Refer to lines 111-125 on page 3 and line 126 on page 4).

- How were volunteers recruited? Where were the volunteers recruited from? How did the authors arrive at this number of 30 volunteers? Was there any calculation to estimate the sample size, or was the study sample estimated by convenience? This information needs to be included in the article, please include it.

> Also, all the experimental procedures and methods were performed in accordance with the relevant guidelines and regulations approved by the IRB-CBUH (Chunbuk National University Hospital) and participants underwent MRI study in Korea Basic Science Institute (KBSI). (Refer to lines 106-109 on page 3).

> The sample was estimated by convenience, and sample size was 30+4 to rule out poor data quality.

(Refer to line 93 on page 3).

- The authors mention that they used two instruments to assess the volunteers: Brief Michigan Alcoholism Screening Test (BMAST) and Alcohol Use Disorder Identification Test-Korea (AUDIT-K), however, they do not provide further information about this assessment, that is, they do not inform how they used this instrument, what these instruments measure, how these measurements are performed (clinical tests, or by questions, functional tests), what are the scores of these instruments? However, they use these data in Table 1, using means and standard deviation. The authors should better explain each of the instruments, explaining how each instrument was used and measures the outcomes, which outcomes are evaluated by these instruments and how the score or punctuation of each of these instruments used by the authors occurs, so that the reader understands better the whole method of this study.

> As your advice, we added the content mentioned above. As follow:

“ BMAST and AUDIT-K, consisted with 10 questions, are self-administered, with individuals responding to the questionnaire. Each question has a scoring system, and the total score is calculated by summing up the scores from each question. Higher scores indicate a higher likelihood of alcohol-related problems or alcohol use disorders. ”

(Refer to lines 99-103 on page 3).

- Item 2.2 MRI Acquisition could have a picture of this acquisition, for a better understanding of how this evaluation was carried out. I suggest including.

> As your advice, we added the figure illustrating sequence for our study from data acquisition to analysis.

Please refer to Figure 1 and Method section (line 90 on page 2).

- Were the parameters used in the MRI acquisition, Brain Volume Difference and FC Analysis based on parameters from previous studies? If yes, include references;

> As your advice, we added reference of our previous study which used same parameters. (In method section, lines 142-143 on page 4)

- The authors used Pearson's Chi-Square test and the Mann-Whitney U-test, however, analyzes using these tests were not mentioned in the statistical analysis section of the article, please include.

> As your recommend, we added sentence about statistical methods in statistical analysis section. (Refer to lines 174-176 on page 4)

> As your advice, we added the contents about Chi square test for handness, and Mann-Whitney U-test for mean age in “ 2.1. Subjects ” section. (Refer to lines 95 and 97-98 on page 3)

> The information about Pearson’s correlation was written at “2.5. Statistical analysis of clinical data and FC” section. (Refer to lines 177-178 on page 4)

- The authors mention that they will make a correlation between (FC intensities and alcoholism severity scores), however, in the methods the authors do not mention which of the instruments used measures the severity of alcoholism, even though these instruments are widely used and known worldwide, the authors need to explain better which instrument was used for which outcome in the methods, so that it makes sense to the readers.

> As your advice, we added the contents about correlation between FC and instruments for alcoholism severity (BMAST and AUDIT-K scores).

Please refer to section of  “2.5. Statistical analysis of clinical data and FC “ (In line 177-178, on page 5).

Results

- Were the ADP volunteers in this study light, moderate or heavy drinkers? This information needs to be clear in the text, not only in this group, but also in the control group.

> Here are some commonly used cutoff scores for the BMAST and AUDIT-K:

Brief Michigan Alcoholism Screening Test (BMAST):

The BMAST does not have a universally agreed-upon cutoff score. However, some studies suggest using a score of 5 or higher as an indicator of potential alcohol-related problems or the need for further assessment. It is important to note that the BMAST is primarily a screening tool, and a higher score does not provide a diagnosis but rather suggests the need for additional evaluation.

In our data, ADPs showed mean score 22.27±5.36.

Alcohol Use Disorder Identification Test-Korea (AUDIT-K):

The AUDIT-K also does not have a single standard cutoff score. Different guidelines and studies may propose varying cutoff scores. However, for identifying possible alcohol use disorders, a total score of 20 or higher is often considered indicative of a potential alcohol use disorder. It is essential to consider the specific guidelines or recommendations being followed when using the AUDIT-K.

In our data, ADPs showed mean score 31.47±7.17.

From these, we think that it is better inform the alcoholism severity just as score because there is no correct and fixed cutoff for alcoholism.

Discussion

- Only in the discussion does the objective of the study appear, however, it seems incomplete, given the many evaluations that the authors carried out. I suggest authors to better delineate the objectives of the study, using terminologies such as: primary objective, secondary objective, tertiary... in this way, the text becomes clearer for readers.

> > As your advice, we wrote the objectives more specifically.

Please refer to Discussion section, in lines 250-256, on page 9.

- In the second paragraph of the discussion, the authors mention that “several studies have highlighted...” and then, they mention only two studies, or the volunteers change the term several, or add more studies, because the term “several studies” from the idea of “many studies” and only two studies seem not to refer to the term used. I suggest modifying.

> As you recommend, we modified the expression form “several studies” to “a couple of studies”.

Please refer to Discussion section, line 264, on page 9. 

- The authors mention some similar studies, however, they do not describe the characteristics of these studies, such as the sample size. Is the sample of these studies larger or smaller than that of the present study? This information can guide the reader as to which results have greater statistical power and also to determine the importance of the present study, when citing studies with a smaller sample size and similar methodology.

> The small sample size of our study is defined based on 30 subjects. This is because a sample size of 30 is considered statistical normality. We added the sentence about “ It is size under 30 that satisfied morality. “. (Refer to Discussion section, line 319 on page 11).  

> A study [41] recruited 30 patients and 30 controls. Maybe the number of 30 is a standard for statistical normality in neuroimaging study.

- The ADP's volunteers were smokers and had a lower level of education, can such outcomes influence the results? What is in the literature about the influence of these two outcomes on the evaluated brain components? I strongly suggest the construction of one or two paragraphs demonstrating the influence of these two outcomes on the evaluated brain characteristics.

> We just considered potential confounding effects of smoking and education levels on the brain structure and function because they are not statistical equal. We only considered statistical difference without any reference.

And we cited the previous papers [40,41] considered were smokers and lower level of education in neuroimaging study. In those previous papers, the participants were smoking and less education level. Nonetheless, the dominance of alcoholism in contributing to differences in brain abnormalities between ADPs and HCs was considered with results.

(Refer to Discussion section, lines 323-327 on page 11).  

- I also missed a paragraph demonstrating a relationship between: what the authors found and its repercussions for the lives of volunteers. Findings in these brain areas can lead to damage in which functional aspects of the volunteers? This information is important, as it can give rise to new research on the subject and also the implementation of intervention programs to benefit the studied population, I strongly suggest the inclusion of this paragraph.

> As your advice, we included the paragraph.

Please refer to Discussion section, lines 312-317 on page 10.

- I also believe that the authors could mention some kind of influence of the chronic use of alcohol on the vestibular system, since they mention the balance problems caused by the continuous use of alcohol on body balance in the introduction. This citation is important, as future studies may include the vestibular components (peripheral and/or central) in future studies on the subject.

> As you recommend, we added the paragraph for vestibular system.

Please refer to Discussion section, lines 291-300 on page 10.

Conclusion

- The authors can leave the conclusion more direct to what the authors found.

> As your advice, we change the conclusion like this :

“ Present study found a significant reduction in WM volume in the GP among male ADPs, and this reduction was associated with changes in FC. These findings suggest that alcoholism can lead to structural and functional network alterations in key brain regions, providing insights into the neural mechanisms underlying ADPs and potential implications for future treatment research. “ (Refer to Conclusion section, in lines 334-344 on page 11)

Reviewer 2 Report

Comments and Suggestions for Authors

This is a very interesting paper investigating regional differences in the white matter and functional changes in the context of patients with alcohol-dependence. The authors explored structural and functional brain abnormalities in male patients and male healthy controls. The paper is well-written and of interest for the journal. However, several changes are recommended before considering it for publication.

ABSTRACT

1- I recommend to better define the study design in the abstract section. Is there any clinical characteristics of the sample to be described in the abstract section?

2- The conclusions should be better separated from the results in the abstract. Perhaps, it would be helpful the terms " The study concluded. Or it can be concluded."

INTRODUCTION

1-I recommend to extend the first paragraph of the introduction by highlighting the social impact of alcohol and the costs in health systems. Medical and social impact of alcohol use disorders should be also emphasized, as well as the impact on mental health (psychosis, psychotic disorders, affective,....).

2- The main aims of the paper should be explained in a separate subsection: 1.1. Aims.

METHODS AND RESULTS

Why have the authors recruited only men? They have included 15 men with alcohol dependence and 15 healthy control's men. Why are women not included?

Is there any difference in the structural and functional analyses of the brain in terms of sex or gender?

DISCUSSION

The different findings from the study should be interpreted with caution with respect to the main bias of the paper. They hava included men. I recommend to add it in the title and the abstract. The title should included the term "men", as well as the abstract should included that the main aims were in "men".

2-How are the education and other sociodemographic characteristics influencing results? It seems that it was found differences in education between groups.

Author Response

ABSTRACT

1- I recommend to better define the study design in the abstract section. Is there any clinical characteristics of the sample to be described in the abstract section?

> we revise the abstract for more precise objective and change the type/design to “Article” from “Original Research”. And corrected the Abstract structure as removing sub-title such as “Objectives”, “Methods” etc.

Refer to Abstract section (on page 1).

> Also, there is clinical characteristic of the sample is that all participants are male, and the clinical differences are brain structural and functional differences with association with clinical characteristics. These meaning is elucidated in revised conclusion in abstract section.

Refer to Abstract section (in lines 29-35 on page 1).

2- The conclusions should be better separated from the results in the abstract. Perhaps, it would be helpful the terms " The study concluded. Or it can be concluded."

> As your suggestion, we revised the conclusion like this: “Based on the observations, it can be concluded that the GP serves as a neural marker that impacts abnormal functional networks in males with ADPs. These findings have important clinical implications, as they provide insights into the neural mechanism underlying the anatomical, functional, and clinical features of alcoholism.”

Refer to Abstract section (in lines 32-35 on page 1).

INTRODUCTION

1-I recommend to extend the first paragraph of the introduction by highlighting the social impact of alcohol and the costs in health systems. Medical and social impact of alcohol use disorders should be also emphasized, as well as the impact on mental health (psychosis, psychotic disorders, affective,....).

> As you recommend, we added related content.

Refer to Abstract section (in lines 51-54 on page 2).

2- The main aims of the paper should be explained in a separate subsection: 1.1. Aims

> We added the sentence of “Hence, the primary objective of our study is to provide a more comprehensive understanding of the neural underpinnings of alcoholism using multi modal approach.” to make our intentions more clearly known in Introduction section (Refer to lines 71-73 on page 2).

> I thought that it is better without separation in Introduction section.

METHODS AND RESULTS

Why have the authors recruited only men? They have included 15 men with alcohol dependence and 15 healthy control's men. Why are women not included?

> Considering your advice, we added the paragraph as follow:

“Women exhibit lower levels of alcohol consumption, initiating their drinking habits at a later stage and displaying lower rates of alcohol dependence in comparison to men [16]. Nevertheless, research has indicated that women experience a more rapid progression from their initial alcohol use to developing alcohol dependence, when compared to men [17]. Consequently, given the presence of gender-related neuroanatomical variances as identified in a recent neuroimaging study [18], exclusively male subjects were chosen to account for these gender distinctions. For this reason, only male subjects were selected in this study.”

Refer to Introduction section (in lines 74-81 on page 2).

Is there any difference in the structural and functional analyses of the brain in terms of sex or gender?

> There is a previous neuroimaging study considering gender difference in aspect of neuronal difference.

> Considering your advice, we added the sentence as follow:

“Consequently, given the presence of gender-related neuroanatomical variances as identified in a recent neuroimaging study [18], exclusively male subjects were chosen to account for these gender distinctions.”

Refer to Introduction section (in lines 78-81 on page 2).

DISCUSSION

The different findings from the study should be interpreted with caution with respect to the main bias of the paper. They hava included men. I recommend to add it in the title and the abstract. The title should included the term "men", as well as the abstract should included that the main aims were in "men".

> Considering your advice, we added the word of “male” in Abstract and Title.

Refer to Title (in lines 2-5 on page 1) and Abstract section (in line 21 and 33 on page 1).

2-How are the education and other sociodemographic characteristics influencing results? It seems that it was found differences in education between groups.

> Here is a paper (1) reported “positive associations between genetically predicted educational attainment and four brain cortical metrics” . We thought that the education level might affect to brain function and/or structure potentially. However to minimize this influence, we considered these factors (education and smoking) as covariates during data processing. (Refer lines323-327 on page 11).

  1. Aida Seyedsalehi and others, Educational attainment, structural brain reserve and Alzheimer’s disease: a Mendelian randomization analysis, Brain, Volume 146, Issue 5, May 2023, Pages 2059–2074, https://doi.org/10.1093/brain/awac392

Round 2

Reviewer 1 Report

Comments and Suggestions for Authors

Congratulations to the authors who did an excellent job correcting the manuscript suggestions. All my requests were fulfilled, so I believe that the article is now clearer for readers and ready to be accepted for publication.